# Biceps Brachii Alterations Following the Latarjet Procedure: A Prospective Multicenter Study

**DOI:** 10.3390/jcm10235487

**Published:** 2021-11-23

**Authors:** Lucca Lacheta, Marco-Christopher Rupp, Andrea Achtnich, Sepp Braun, Mark Tauber, Andreas B. Imhoff, Peter Habermeyer, Frank Martetschläger

**Affiliations:** 1Department of Orthopaedic Sports Medicine, Technical University of Munich, Ismaninger Strasse 22, 81675 Munich, Germany; marco.rupp@tum.de (M.-C.R.); andrea.achtnich@tum.de (A.A.); s.braun@gelenkpunkt.com (S.B.); imhoff@tum.de (A.B.I.); frank.martetschlaeger@atos.de (F.M.); 2Center for Musculoskeletal Surgery, Charitè Universitaetsmedizin Berlin, Campus Virchow, 13353 Berlin, Germany; 3Gelenkpunkt—Sports and Joint Surgery Innsbruck, Olympiastrasse 39, 6020 Innsbruck, Austria; 4Department of Shoulder and Elbow Surgery, ATOS Clinic Munich, Effnerstrasse 38, 81925 Munich, Germany; mark.tauber@atos.de (M.T.); peter.habermeyer@atos.de (P.H.); 5Department of Orthopaedics and Traumatology, Paracelsus Medical University, 5020 Salzburg, Austria

**Keywords:** biceps brachii, Latarjet procedure, strength, popeye deformity, shoulder instability

## Abstract

Purpose: To prospectively investigate the postoperative forearm supination and elbow flexion strength of both upper extremities and popeye deformity in patients who underwent a mini-open Latarjet procedure for anterior shoulder instability. Methods: Patients who underwent a mini-open Latarjet procedure at two specialized shoulder centers were prospectively evaluated preoperatively (T0) and at least 6 months (T1) after surgery. Subjects were tested for elbow flexion and forearm supination strength of both upper extremities using an isometric dynamometer and customized torque dynamometer. Clinical outcome was assessed by the Constant Score (CS), American Shoulder and Elbow Score (ASES) and Simple Shoulder test (SST). Popeye deformity was defined as a distalization of the greatest circumference of the biceps muscle belly towards the lateral epicondyle of the elbow. Results: A total of 20 patients with a mean age of 27 ± 6 years were included in the study. At a mean follow-up of 10 ± 3 months, the elbow flexion strength was restored to the preoperative state (*p* = 0.240). Forearm supination strength significantly decreased at final follow-up, to 88 % in the surgical arm (*p* = 0.015) vs. 90 % in the non-surgical arm (*p* = 0.023). There was no statistical difference when comparing both arms concerning elbow flexion strength (*p* = 0.510) and forearm supination strength (*p* = 0.495). No significant popeye deformity was observed in both arms (*p* = 0.111 vs. *p* = 0.508). Clinical outcome scores improved significantly from 73 ± 18 to 82 ± 13 (*p* = 0.014) for CS and 76 ± 22 to 89 ± 12 (*p* = 0.008) for ASES score preoperatively to final follow-up. No difference in the SST was documented (*p* = 0.10). Conclusion: The Latarjet procedure showed to preserve elbow flexion strength and provided comparable forearm supination strength compared to the uninjured arm with reliable clinical outcome in this study population. However, a decrease of forearm supination strength in both arms was persistent at a mean of 10 months postoperatively. No popeye deformity was noted in the postoperative examinations. Level of evidence: Case series, Level III.

## 1. Introduction

Anterior shoulder instability mostly affects a young and active population [1,2]. Young patient age, presence of a bony lesion (glenoid or humeral side), failed soft tissue stabilization (Bankart repair) as well as an active lifestyle have shown to be risk factors for recurrent instability and therefore are a relative indication for bony augmentation [3,4,5,6].

One commonly described surgical technique for bony augmentation is the Latarjet procedure, involving a transfer of the coracoid process to the anterior glenoid with its attached conjoint tendons (M. biceps brachii and M. coracobrachialis) through a subscapularis muscle split [7]. This open or arthroscopic technique can be used as a salvage procedure for failed primary soft tissue stabilization, as a first-line treatment in the case of bony defects or for patients at high risk [8].

Previously published early outcome reports are satisfactory, with less pain and fast recovery. The recurrence and revision rates were low, up to 6.5% in the studied population [9]. This is consistent with recently reported mid-term results, with a recurrent instability rate of 1.6% and good clinical outcome [3]. Both the open and arthroscopic Latarjet procedures provide significant improvements in function and outcome results, with similar low rates of recurrent instability and complications [10].

However, the Latarjet procedure is a non-anatomic shoulder reconstruction procedure. Due to a transfer of the biceps brachii and coracobrachialis muscle attachment to the anterior glenoid, this procedure might modify the shoulder biomechanics, not only concerning stability. The influence of the inherent tendon transfer during the Latarjet procedure on function and morphology of the biceps brachii muscle is yet unknown.

Therefore, the purpose of this study was to evaluate forearm supination and elbow flexion strength of the upper extremity and popeye deformity in patients who have had the mini-open Latarjet procedure using the classic technique to treat anterior shoulder instability. It was hypothesized that patients would achieve full forearm supination and elbow flexion strength compared to the contralateral arm at short-term follow-up.

## 2. Methods

### 2.1. Study Population

This was an Institutional Review Board–approved level III prospective study. Over a 2-year period (2016–2018), 20 patients were enrolled in this study if they were 18 years or older at the time of surgery, were indicated for Latarjet procedure due to anterior shoulder instability and had an unsuspicious course of the long head of the biceps tendon in magnetic resonance imaging (MRI) and clinical examination.

Patients were excluded if they suffered from biceps brachii pathologies with positive clinical signs for the long head of the biceps (Palm-up test, Speed test, O’Brien test), had pathologic imaging findings for the long head of the biceps tendon, had any previous surgical long head of the biceps tenodesis/tenotomy, had elbow or wrist surgery on the affected side or were unable or not willing to give informed consent. 

Patient demographics, surgical data and clinical outcome parameters were collected prospectively and reviewed retrospectively. Preoperatively and at the time of follow-up (at least six months after surgery), patients completed strength testing, measurement of popeye deformity and outcome evaluations, including the Constant Murley Score (CS), American Shoulder and Elbow Score (ASES) and Simple Shoulder Test (SST).

Achievement of full forearm flexion and supination strength compared to the contralateral arm was defined as the endpoint of the study.

### 2.2. Subject Preparation

After identification during an outpatient clinic visit, either the senior physician or attending fellow asked patients to participate as volunteers for this study. Once they gave their written consent to participate and read through the study details, the subjects were further introduced to and educated in the testing equipment on the day of admission for surgery. All testing was conducted by a single independent examiner, who was not involved in surgical treatment, at two different centers and lasted approximately 30 min.

Before testing, subjects were instructed how to position and use the test equipment. Additional time was given to familiarize them with the test setting.

### 2.3. Strength Testing

Elbow flexion strength and forearm supination strength were measured by the use of an Isobex isometric dynamometer (Cursor, Bern, Switzerland) (Figure 1A) and digital torque peak adapter (BGS Technic KG, Wermelskirchen, Germany) (Figure 1B) with custom-made fixation according to similar designs or other investigators used before [11,12,13]. Testing was conducted preoperatively on the day of surgery and at the time of follow up, at least 6 months postoperatively.

Forearm Supination Testing: Subjects were instructed to sit on the testing chair, with their upper arm applied to the upper body. Their shoulder was placed in adduction and neutral external/internal rotation, and the elbow was flexed in 90°. The start position of the forearm was in the neutral position with respect to supination/pronation (Figure 2). The subjects were allowed to practice as long as needed to become comfortable with the test situation and were free to choose which arm to start with. The subjects were instructed to supinate the forearm with maximum strength, and peak torque was recorded. Testing consisted of two alternating repetitions. Subjects paused for 10 s before changing arms and 2 min before second repetition. The subjects were constantly checked and corrected by the examiner for correct positioning prior to testing to avoid positioning bias.

Elbow Flexion Testing: The same position was used for elbow flexion testing, with the forearm in 90° supination and the elbow in 90° flexion (Figure 3). Subjects were first given several trials to familiarize with the new test situation, without the use of maximum strength to avoid fatigue of the biceps muscle. Patients were instructed to flex the elbow with maximum strength, and peak strength was noted. Again, testing consisted of two alternating repetitions, which were paused for 10 s before changing arms and 2 min before second repetition.

### 2.4. Popeye Deformity

To objectify a distalization of the biceps brachii muscle (popeye deformity), patients were asked to flex their elbow to 90° and fully supinate the forearm (to have a maximally contracted muscle). The largest circumference of the muscle belly of the upper arm (Figure 4A) and its distance to the lateral epicondyle of the elbow (Figure 4B) were measured for both arms preoperatively and at the time of follow-up. A distalization of the largest circumference of more than one centimeter was stated as positive for popeye deformity.

### 2.5. Clinical Outcome Assessment

All patients underwent clinical assessments prior to the Latarjet procedure and at final follow-up evaluation. The functional results were assessed using the Constant Score (CS), the American Shoulder and Elbow Score (ASES) and the Simple Shoulder Test (SST). 

### 2.6. Surgical Procedure

All patients were treated with the mini-open Latarjet procedure, using the classic technique as previously described by Young and Walch [14].

For postoperative management, all patients had their arm secured in a sling until weeks 4–6. Range of motion was limited to 45° of abduction and flexion, with external rotation to 0°, for a duration of 3 weeks. Abduction and flexion were increased to 90° in week 4. By the beginning of week 7, range of motion was unlimited. Patients were allowed to start strengthening and return to sports-related training after week 12. High-risk sports (e.g., contact sport) and competition was suspended for 5 months postoperatively.

### 2.7. Statistical Analysis

The primary goal of this study was to investigate if the Latarjet procedure leads to a relevant decrease in flexion and supination strength of the elbow. A power analysis and sample size calculation was performed prior to testing. Assuming an actual probability of success of 95%, 20 patients must be included in the study in order to be able to reject the null hypothesis with a probability (=power) of 80%.

All statistical analyses were performed using SPSS version 11.0 (SPSS, Chicago, IL, USA). The pre- and postoperative strength measurements and outcome scores of the study population were compared with a Wilcoxon signed-rank test. All results are presented as means and standard deviation unless otherwise stated.

## 3. Results

Between 2016 and 2018, 20 patients with a mean age of 27.2 ± 6.1 years were included in the study and eligible for follow up (100%) at a mean follow-up time of 9.8 ± 3.3 months. In this time period, no recurrent anterior shoulder instability occurred, and no patients underwent further surgery. Patients’ demographics are listed in Table 1.

### 3.1. Within-Arm Comparison

The surgical arm treated with the Latarjet procedure showed no change of flexion strength from the preoperative state to final follow-up (*p* = 0.240). Forearm supination strength decreased to 88 % of forearm supination strength with statistical significance at final follow-up (*p* = 0.015).

In comparison, the non-surgical arm showed similar changes in flexion strength, with an increase of flexion strength to a mean of 110% preoperatively to T1 (*p* = 0.147) and a significant decrease of forearm supination strength to 90 % at T1 (*p* = 0.023). All strength measurements with means and standard deviation are summarized in Table 2.

No popeye deformity was observed in both arms (Table 3) at final follow-up. The distance from greatest circumference of the muscle belly of the upper arm to the lateral epicondyle of the elbow stayed constant in the surgical arm and the non-surgical arm (*p* = 0.111 vs. *p* = 0.508) preoperatively to the time of final follow-up.

### 3.2. Between-Arm Comparison

When comparing across both arms, we found no significant differences in forearm supination strength (*p* = 0.495) nor in elbow flexion strength (*p* = 0.510) at the time of final follow-up. There were also no significant differences between the surgical arm and non-surgical arm concerning popeye deformity (*p* = 1.000).

### 3.3. Clinical Outcome

With regard to shoulder scores, clinical outcome had significantly improved in CS (*p* = 0.014) and ASES (*p* = 0.008) score preoperatively to final follow-up; SST showed no significant changes (*p* = 0.10). All functional results are summarized in Table 4.

## 4. Discussion

The main finding of this study is the preservation of elbow flexion strength and the excellent postoperative clinical outcome in patients treated with the Latarjet procedure at a mean follow-up of 10 months. Although we found comparable results for forearm flexion strength compared to the uninjured, healthy arm, both arms showed a significant decrease of forearm supination strength over time compared to the preoperative state.

To date, only few studies focus on clinical outcomes and return to activity following the Latarjet procedure [1,3,8,15]. Changes on biceps brachii strength and muscle belly deformity are unknown.

There is only one study that examined the biceps brachii muscle after the Bristow-Latarjet procedure using sonographic changes. Castoldi et al. retrospectively compared changes in muscle morphology and size by ultrasound in 26 patients who underwent the Bristow-Latarjet procedure, with a mean follow-up time of 61.3 months (range 12–108), to 23 healthy volunteers. They found no modification in size or morphology in the Bristow-Latarjet group compared to the healthy group [16]. This is in line with this study’s findings, showing no changes in muscle belly presentation (no popeye indications). Additionally, constant muscle strength for elbow flexion strength over time suggests that no major morphological differences occurred. 

The main functions of the biceps muscle are elbow flexion and forearm supination. Several investigators established elbow flexion strength and forearm supination strength testing to evaluate the biceps function in patients suffering from pathologies of the proximal long head of the biceps or distal biceps tendon [17,18]. No reports exist on strength testing regarding affection of the short head of the biceps muscle following a transfer of the conjoint tendons. Therefore, this study’s results are difficult to compare to the existing literature.

This study showed that the Latarjet procedure, with its non-anatomical “tenodesis” of the short head of the biceps, achieved improved elbow flexion strength compared to the preoperative state. Despite these satisfying results for elbow flexion strength, we noticed a significant postoperative reduction of forearm supination strength on the surgical side to 88% and to 90% on the healthy side.

The increase of flexion strength may be a result of constant postoperative rehabilitation and training, but also might be due to the newly gained mental trust by the treated glenohumeral instability of the shoulder. The decrease of forearm supination strength showed no statistical difference when comparing both arms. We interpret the findings not as any debilitating effect of the Latarjet procedure on the biceps muscle. In contrast, this effect may be the result of the postoperative rehabilitation program and patient activity profile following the Latarjet procedure, which does not focus on early strengthening of the biceps muscle or supination strength. This could also explain the absence of side differences. Furthermore, with a mean age of 27 years at time of surgery, a modification of sport participation towards less resilient activities postoperatively may be conceivable in this study population. Overall, the finding of decreased forearm supination strength has no influence on subjective well-being of the patients, which is reflected in the excellent clinical scores.

Further investigations with larger cohorts and longer follow-up intervals may help to confirm these observations.

While this study could present interesting findings in this series, limitations are acknowledged.

First, we studied values of peak strength and not an “exercise to muscle fatigue”. Second, relatively low numbers of patients were used; therefore, we provided a power analysis and calculated sample size. This gives the reader a sense of what sample sizes are required to make the noted comparison significant. Third, all patients were given a postoperative rehabilitation protocol; however, it is not known whether it was strictly followed, as physical therapy took place in an outpatient setup.

## 5. Conclusions

The Latarjet procedure showed to preserve elbow flexion strength and provided comparable forearm supination strength compared to the uninjured arm with reliable clinical outcome in this study population. However, a decrease of forearm supination strength in both arms was persistent at a mean of 10 months postoperatively. No popeye deformity was noted in the postoperative examinations.

## Figures and Tables

**Figure 1 jcm-10-05487-f001:**
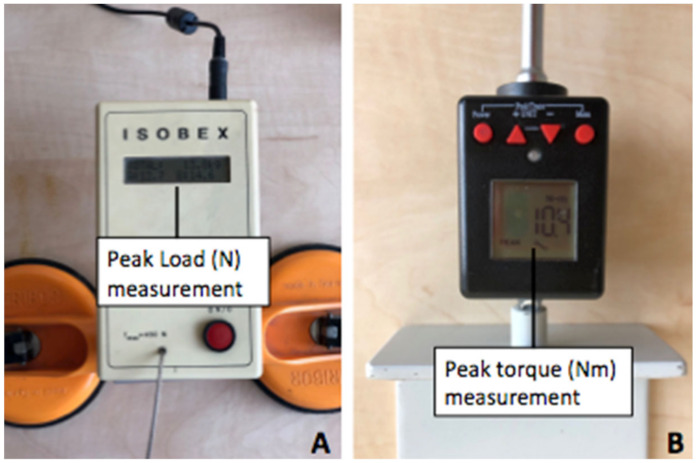
Isobex isometric dynamometer (Cursor, Bern, Switzerland) for peak load (N) measurement of elbow flexion strength (**A**); Digital torque adapter (BGS technic KG, Wermelskirchen, Germany) with custom-made fixation for peak torque (Nm) measurement (**B**).

**Figure 2 jcm-10-05487-f002:**
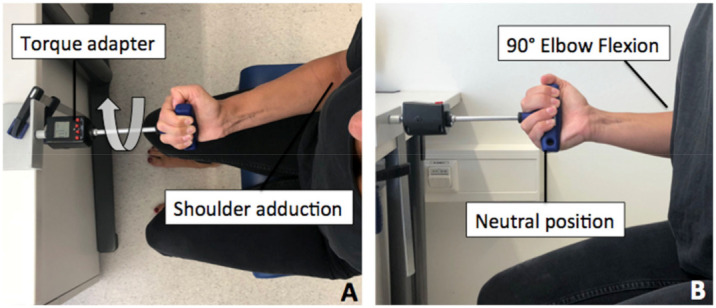
Start position with fully adducted shoulder (**A**), elbow in 90° flexion and neutral position with respect to supination/pronation (**B**). Patient was asked to perform maximum supination (arrow).

**Figure 3 jcm-10-05487-f003:**
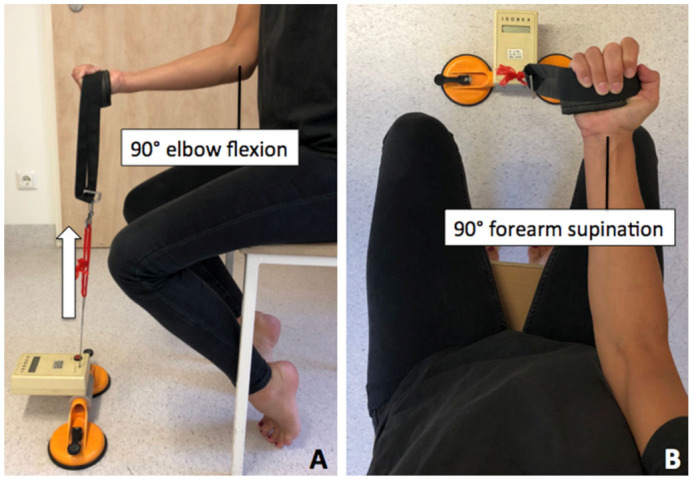
Patient positioned in upright position with 90° elbow flexion (**A**) and forearm supination to 90° (**B**) with perpendicular force vector (arrow).

**Figure 4 jcm-10-05487-f004:**
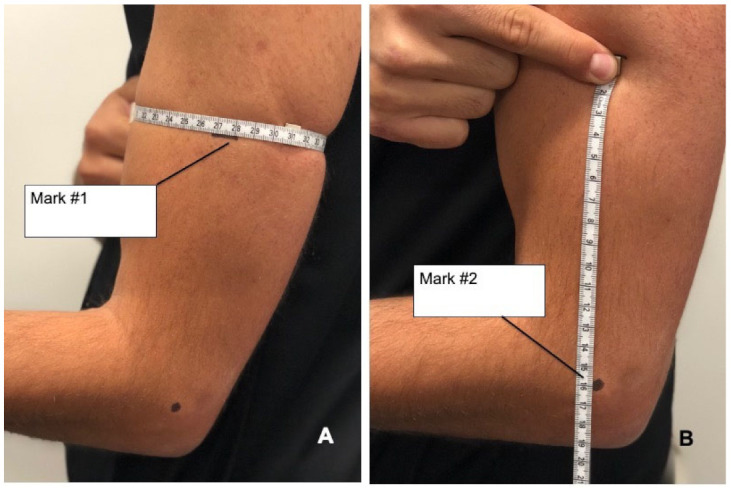
(**A**) Measurement and marking (Mark #1) of the height of the largest circumference of the upper arm with forearm in full supination. (**B**) Distance from the height of the largest circumference (Mark #1) to the lateral epicondyle of the elbow (Mark #2).

**Table 1 jcm-10-05487-t001:** Patient’s demographics.

Parameter	Number (%)
Male/Female	16 (80)/4 (20)
Right/left shoulder	15 (75)/5 (25)
Dominant/Non-dominant arm	14 (70)/6 (30)
Prior surgery/No prior surgery	9 (45)/11 (55)

**Table 2 jcm-10-05487-t002:** Mean values, standard deviation and *p*-values (* indicates statistical significance) for elbow flexion and forearm supination strength measurements (in percent) compared to preoperative baseline measurements.

	Surgical Arm		Non-Surgical Arm	
	T1	*p*-Value	T1	*p*-Value
Peak Flexion Strength (N)	103.0 ± 28.2%	0.240	109.2 ± 26.1%	0.147
Peak Supination Torque (Nm)	87.6 ± 34.6%	0.015 *	89.9 ± 32.5%	0.023 *

**Table 3 jcm-10-05487-t003:** Mean values for upper arm circumferences and distance to the lateral epicondyle.

	Surgical Arm			Non-Surgical Arm		
	Preoperative	Postoperative	*p*-Value	Preoperative	Postoperative	*p*-Value
Distance Circumference to Lateral Epicondyle (cm)	12.5 ± 1.1	12.9 ± 1.1	0.111	13.7 ± 5.1	12.9 ± 1.1	0.508

**Table 4 jcm-10-05487-t004:** Mean values for functional outcome scores (* indicates statistical significance).

Outcome Score	Time	Result	*p*-Value
Constant Score	preoperative	72.6 ± 17.8	0.014 *
	postoperative	82.2 ± 12.9	
ASES Score	preoperative	75.6 ± 22.1	0.008 *
	postoperative	89.4 ± 12.1	
SST	preoperative	10.3 ± 1.5	0.10
	postoperative	11.3 ± 1.3	

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
