# Peer review of "Biceps Brachii Alterations Following the Latarjet Procedure: A Prospective Multicenter Study"

_jcm, 2021, doi:10.3390/jcm10235487_

Round 1

Reviewer 1 Report

over all good and important topic

limitation- low number of patients

methods- 2.3 strengthening testing at the end of the paragraph - I believe it should be ".....at least 6 months postoperatively"

Author Response

over all good and important topic

Author's response: Thank you for your kind words. 

limitation- low number of patients

Author's response: Thank you for your comment. We provided a power analysis to detect significant differences. 

methods- 2.3 strengthening testing at the end of the paragraph - I believe it should be ".....at least 6 months postoperatively"

Author's response: Thank your for your comment. You are correct, the manuscript was changed accordingly. 

Reviewer 2 Report

The authors presented very interesting results with respect to the functional and morphological changes in the biceps brachii after the Latarjet procedure. The manuscript was well written. Methods were described in detail, and the results were clear and easy to understand. The discussion was well documented and clearly relevant to the conclusions. Thus, the reviewer will suggest some minor corrections.

Abstract

In lines 32-33, the authors described, “No difference in the SST was documented (p=0.010)”. However, the p-value in this sentence was less than the general statistical significant value (0.05). Would you please clarify this point? Moreover, a similar description can be seen in line 195 and Table 4.

Methods

In line 93, the authors used the term “und.” The reviewer thinks that the word “and” may be correct in this context. Would you please reconsider the sentence?

Results

Results were presented with appropriate figures and tables and were easy to understand. Please clarify the contradiction in line 195 and Table 4.

Discussion

The discussion was well documented and relevant to the conclusions.

Author Response

The authors presented very interesting results with respect to the functional and morphological changes in the biceps brachii after the Latarjet procedure. The manuscript was well written. Methods were described in detail, and the results were clear and easy to understand. The discussion was well documented and clearly relevant to the conclusions. Thus, the reviewer will suggest some minor corrections.

Authors's response: Thank you for your kind words. 

Abstract

In lines 32-33, the authors described, “No difference in the SST was documented (p=0.010)”. However, the p-value in this sentence was less than the general statistical significant value (0.05). Would you please clarify this point? Moreover, a similar description can be seen in line 195 and Table 4.

Authors's response: Thank you for your critical remark. This was a typo, the correct number is p=0.10 (not significant). We apologize for that. Text and table were changed accordingly. 

Methods

In line 93, the authors used the term “und.” The reviewer thinks that the word “and” may be correct in this context. Would you please reconsider the sentence?

Authors's response: Thank you for your comment. You are correct, it now reads "and". 

Results

Results were presented with appropriate figures and tables and were easy to understand. Please clarify the contradiction in line 195 and Table 4.

Authors's response: Thank you for your kind words. Please see the comment aboce, table 4 was changed accordingly. 

Discussion

The discussion was well documented and relevant to the conclu

Authors's response: Thank you for your words and time reviewing our manuscript!